# Non-reversible Gaussian processes for identifying latent dynamical structure in neural data

**Virginia M. S. Rutten**
Gatsby Computational Neuroscience Unit
University College London, London, UK
& Janelia Research Campus, HHMI
Ashburn, VA, USA
ruttenv@janelia.hhmi.org

**Alberto Bernacchia**
MediaTek Research
Cambourne Business Park
Cambridge, UK
alberto.bernacchia@mtkresearch.com

**Maneesh Sahani**
Gatsby Computational Neuroscience Unit
University College London
London, UK
maneesh@gatsby.ucl.ac.uk

**Guillaume Hennequin**
Department of Engineering
University of Cambridge
Cambridge, UK
g.hennequin@eng.cam.ac.uk

## Abstract

A common goal in the analysis of neural data is to compress large population recordings into sets of interpretable, low-dimensional latent trajectories. This problem can be approached using Gaussian process (GP)-based methods which provide uncertainty quantification and principled model selection. However, standard GP priors do not distinguish between underlying dynamical processes and other forms of temporal autocorrelation. Here, we propose a new family of "dynamical" priors over trajectories, in the form of GP covariance functions that express a property shared by most dynamical systems: temporal non-reversibility. Non-reversibility is a universal signature of autonomous dynamical systems whose state trajectories follow consistent flow fields, such that any observed trajectory could not occur in reverse. Our new multi-output GP kernels can be used as drop-in replacements for standard kernels in multivariate regression, but also in latent variable models such as Gaussian process factor analysis (GPFA). We therefore introduce GPFADS (Gaussian Process Factor Analysis with Dynamical Structure), which models single-trial neural population activity using low-dimensional, non-reversible latent processes. Unlike previously proposed non-reversible multi-output kernels, ours admits a Kronecker factorization enabling fast and memory-efficient learning and inference. We apply GPFADS to synthetic data and show that it correctly recovers ground truth phase portraits. GPFADS also provides a probabilistic generalization of jPCA, a method originally developed for identifying latent rotational dynamics in neural data. When applied to monkey M1 neural recordings, GPFADS discovers latent trajectories with strong dynamical structure in the form of rotations.

## 1   Introduction

The brain has evolved as a rich dynamical system to control and coordinate the other dynamical systems that make up the body. High-dimensional neural activity can often be efficiently recapitulated by lower dimensional latent dynamics, and multiple methods have been proposed over the years to tackle the challenge of extracting interpretable and actionable latent trajectories.

A first class of methods focuses on explicitly learning the transition function of an underlying dynamical system. These include parametric models such as linear dynamical systems (LDS) models (Buesing et al., 2012a,b; Churchland et al., 2012; Macke et al., 2011; Roweis and Ghahramani, 1999) and switching variants (Linderman et al., 2017; Petreska et al., 2011), probabilistic deep learning approaches such as LFADS (Pandarinath et al., 2018), as well as more flexible non-parametric models of the transition function and its uncertainty (Deisenroth and Rasmussen, 2011; Duncker et al., 2019). While appealing in principle, the latter methods do not allow exact inference, must combat pervasive local optima during training, and are computationally intensive. As such, they have yet to be more widely adopted in the field.

The second class of methods focuses on modeling the statistics of the latent processes directly, rather than learning a dynamical model for them. Such methods include Gaussian-process factor analysis (GPFA) and variants (Yu et al., 2009). Gaussian process (GP)-based methods are data efficient and have closed form formulas allowing for uncertainty estimation and principled model selection (Rasmussen and Williams, 2006). Yet, these models fail to capture features of dynamical systems beyond basic smoothness properties, limiting our capacity to study the dynamics of brain computations.

We set out to bridge these two classes of models by imparting some notion of "dynamics" to GP-based models. A key property of autonomous dynamical systems is that they define a consistent mean flow field in state space, such that any segment of state-trajectory produced by the system is unlikely to be visited in the opposite direction (though this is not true of strongly input-driven, or partially observed systems). To capture this property in the Gaussian process framework, we introduce a measure of second-order non-reversibility and derive a new family of GP covariance functions for which this measure can be made arbitrarily large. These kernels can be derived from a variety of usual scalar stationary covariance functions, such as the squared-exponential kernel or the more expressive spectral mixture kernel (Wilson and Adams, 2013). Conveniently, our non-reversible multi-output GP construction affords a specific Kronecker structure; we discuss how this property enables scalability to very large datasets. We validate these kernels on a regression problem where we show that non-reversible covariances yield better model fits than reversible ones for datasets originating from dynamical systems. We then introduce non-reversible kernels in GPFA, and call this variant GPFADS, Gaussian Process Factor Analysis with Dynamical Structure. We show how GPFADS allows demixing of dynamical processes from other high-variance latent distractors, even where demixing could not be performed by comparing lengthscales alone. Finally, we apply GPFADS to population recordings in monkey primary motor cortex. We find that it discovers latent processes with clear rotational structure, consistent with earlier findings (Churchland et al., 2012).

## 2 Background: Gaussian Process Factor Analysis (GPFA)

**Notation** In the following, we use bold $\boldsymbol{x}$ for column vectors, and capital $X$ for matrices whose elements we denote by $x_{ij}$. In any context where matrix $X$ has been introduced, $\tilde{\boldsymbol{x}}$ is a shorthand notation for $\text{vec}(X^\top)$, where $\text{vec}(\cdot)$ is the operator which vertically stacks the columns of the matrix. The transpose is needed for consistency with the convention used in the rest of the paper, which requires that the rows be transposed and stacked vertically instead of columns. Finally, $I_N$ denotes the $N \times N$ identity matrix, and $\mathbf{1}_N$ denotes the column vector whose $N$ elements are all ones.

Latent variable models offer a parsimonious way of capturing statistical dependencies in multivariate time series. Gaussian process factor analysis (GPFA; Yu et al., 2009) is one such popular model used for simultaneous dimensionality reduction and denoising/smoothing of neural population recordings. Missing data are straightforward to handle, but for simplicity of exposition, we assume that observations $\boldsymbol{y}(t) \in \mathbb{R}^N$ are available for each of $N$ variates at each of $T$ time points. GPFA assumes that such observations arise as the noisy linear combination of a smaller set of $M$ latent trajectories, $\boldsymbol{x}(t) = (x_1(t), \ldots, x_M(t))^\top \in \mathbb{R}^M$, each modelled as an independent Gaussian process. Formally,

$$x_i(\cdot) \sim \mathcal{GP}(0, k_i(\cdot, \cdot))$$
$$\boldsymbol{y}(t) \sim \mathcal{N}(\boldsymbol{\mu} + C\boldsymbol{x}(t), R) \tag{1}$$

where $k_i(\cdot, \cdot)$ is the covariance function (or "kernel") of the $i^{\text{th}}$ latent GP. The model is trained by maximizing the log marginal likelihood $\mathcal{L}(\boldsymbol{\theta})$ w.r.t. the parameter vector $\boldsymbol{\theta}$, which comprises all kernel parameters (see below), a mean vector $\boldsymbol{\mu} \in \mathbb{R}^{N \times 1}$, a mixing matrix $C \in \mathbb{R}^{N \times M}$, and a diagonal

matrix of private observation noise variances $R \in \mathbb{R}^{N \times N}$. For a data sample $Y \in \mathbb{R}^{N \times T}$, the log marginal likelihood is proportional (up to an additive constant) to

$$\mathcal{L}(\theta, Y) \propto -\log|K_{yy}| - [\tilde{\boldsymbol{y}} - \boldsymbol{\mu} \otimes \mathbf{1}_T]^\top K_{yy}^{-1} [\tilde{\boldsymbol{y}} - \boldsymbol{\mu} \otimes \mathbf{1}_T] \tag{2}$$

$$\text{with} \quad K_{yy} = (C \otimes I_T)K_{xx}(C^\top \otimes I_T) + (R \otimes I_T) \tag{3}$$

where $\tilde{\boldsymbol{y}} = \text{vec}(Y^T)$, $K_{xx} \in \mathbb{R}^{MT \times MT}$ is the prior Gram matrix, and $\otimes$ denotes the Kronecker product. As the latents are a priori independent in this original formulation, $K_{xx}$ is block diagonal with the $i^{\text{th}}$ diagonal block corresponding to the $T \times T$ Gram matrix of latent $x_i$.

Given a particular observation $\tilde{\boldsymbol{y}} \in \mathbb{R}^{NT}$, the posterior mean and covariance over latent trajectories are given by:

$$\mathbb{E}(\tilde{\boldsymbol{x}}|\tilde{\boldsymbol{y}}) = K_{xx}(C^\top \otimes I_T)K_{yy}^{-1}[\tilde{\boldsymbol{y}} - \boldsymbol{\mu} \otimes \mathbf{1}_T] \tag{4}$$

$$\text{Cov}(\tilde{\boldsymbol{x}}|\tilde{\boldsymbol{y}}) = K_{xx} - K_{xx}(C^\top \otimes I_T)K_{yy}^{-1}(C \otimes I_T)K_{xx}. \tag{5}$$

Note that once $\tilde{\boldsymbol{v}} = K_{yy}^{-1}[\tilde{\boldsymbol{y}} - \boldsymbol{\mu} \otimes \mathbf{1}_T]$ is computed, the rest of the computation of the posterior mean can be sped up by using the Kronecker identity $(C^\top \otimes I_T)\tilde{\boldsymbol{v}} = \text{vec}(V^\top C)$. We further outline how the relevant quantities for inference and learning can be stably and efficiently computed for the original and our own model in Appendix F.1. We also discuss a highly scalable implementation in Appendix F.2.

# 3 Nonreversible Gaussian processes

A major limitation of the original GPFA model summarized in Section 2 is the assumption that the latent processes are independent *a priori*. This in turn severely impairs the ability to extrapolate or look into any learned prior relationships between latents in search for dynamical structure (e.g. consistent phase lags between latents, delays, etc.).

In this paper, we introduce novel multi-output covariance functions aimed at expressing a key property of dynamical systems: that they produce state-space trajectories that follow lawful flow fields and are therefore temporally non-reversible. We begin by formalizing this idea of temporal non-reversibility for stationary GPs, before describing our construction of non-reversible multi-output GP kernels, which we then combine with GPFA, yielding GPFADS, Gaussian Process Factor Analysis with Dynamical Structure.

## 3.1 Quantifying non-reversibility and decomposing multi-output GP covariances

Consider a stationary zero-mean multi-output Gaussian process $\boldsymbol{x}(t) = (x_1(t), \ldots, x_M(t))$ with covariance functions $k_{ij}(\tau) \triangleq \mathbb{E}[x_i(t)x_j(t+\tau)]$. We define $\boldsymbol{x}$ to be temporally reversible if, and only if, all pairwise cross-covariance functions have no odd part, i.e. $k_{ij}(\tau) = k_{ij}(-\tau)$ for all $i \neq j$ and $\tau \in \mathbb{R}$. This is equivalent to the condition that the spatial cross-covariance matrix $K(\tau) \triangleq \mathbb{E}[\boldsymbol{x}(t)\boldsymbol{x}(t+\tau)^\top]$ be symmetric for any lag $\tau$. Thus, only multi-output GPs can be made non-reversible. To quantify departure from pure reversibility in a multi-output GP, we introduce the following measure of non-reversibility:

$$\zeta = \left( \frac{\int_{-\infty}^{\infty} \|K(\tau) - K(-\tau)\|_{\text{F}}^2 \, d\tau}{\int_{-\infty}^{\infty} \|K(\tau) + K(-\tau)\|_{\text{F}}^2 \, d\tau} \right)^{1/2} \tag{6}$$

where $\|\cdot\|_{\text{F}}$ denotes the Frobenius norm. In Appendix A, we prove that $0 \leq \zeta \leq 1$. We note that, by this definition, any scalar (one-dimensional) GP is necessarily fully reversible ($\zeta = 0$).

Our goal is to construct GP covariance functions that break reversibility. As a first step, we prove in Appendix B that any stationary $M$-output GP covariance admits a finite "Kronecker" decomposition:

$$K(\tau) = \sum_{\ell=1}^{n^+} \lambda_\ell^+ A_\ell^+ f_\ell^+(\tau) + \sum_{\ell=1}^{n^-} \lambda_\ell^- A_\ell^- f_\ell^-(\tau) \quad \text{with} \quad \begin{cases} \text{Tr}\left(A_\ell^\pm A_{\ell'}^{\pm \top}\right) = \delta_{\ell\ell'} \\ \int f_\ell^\pm(\tau)f_{\ell'}^\pm(\tau) \, d\tau = \delta_{\ell\ell'} \end{cases} \tag{7}$$

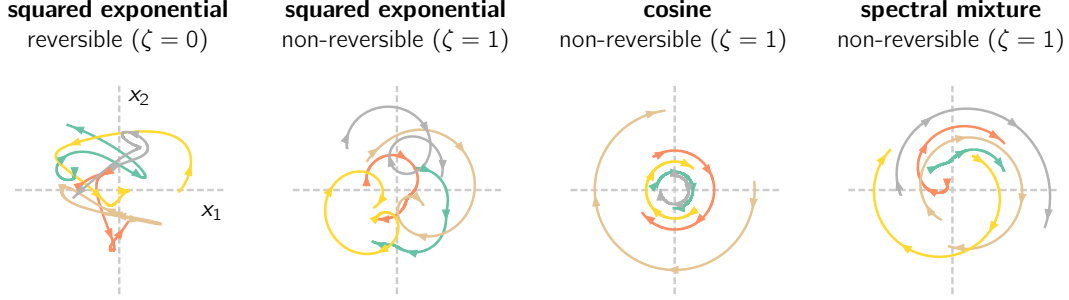

**squared exponential**
reversible ($\zeta = 0$)

**squared exponential**
non-reversible ($\zeta = 1$)

**cosine**
non-reversible ($\zeta = 1$)

**spectral mixture**
non-reversible ($\zeta = 1$)

Figure 1: **Sample trajectories from various planar GP kernels,** defined in Eq. 9. Five samples trajectories are shown over an interval of 5 time units for the following kernels, in this order: SE kernel $f(\tau) = \exp(-\tau^2/2)$ with $\alpha = 0$, SE kernel with $\alpha = 1$, cosine kernel $f(\tau) = \cos(0.15 \times 2\pi\tau)$ with $\alpha = 1$, and spectral mixture kernel $f(\tau) = \exp(-\tau^2/(2 \times 2.5^2))\cos(0.06 \times 2\pi\tau)$ with $\alpha = 1$. Here we have set $\sigma_1 = \sigma_2$ and $\rho = 0$, resulting in spherical planar processes.

where $n^+ + n^- = M^2$ and $\lambda_\ell^\pm \geq 0$. In this decomposition, $\{A_\ell^\pm\}$ is a collection of $M \times M$ symmetric (+) or skew-symmetric (-) matrices, all orthonormal to each other in the sense expressed in Eq. 7. Similarly, $\{f_\ell^\pm\}$ is a matching set of orthonormal even (+) or odd (-) scalar functions. We assume without loss of generality that the weighting coefficients $\{\lambda_\ell^\pm\}$ are ordered by decreasing value within each (+) and (-) sets.

The decomposition in Eq. 7 is a "Kronecker" decomposition (Van Loan, 2000), because the Gram matrix instantiating $K(\tau)$ at a discrete set of time points is composed of a sum of Kronecker products: $K = \sum_\ell \lambda_\ell^+ A_\ell^+ \otimes F_\ell^+ + \sum_\ell \lambda_\ell^- A_\ell^- \otimes F_\ell^-$ with $F_\ell^+$ and $F_\ell^- \in \mathbb{R}^{T \times T}$. This decomposition conveniently isolates terms that either strengthen (+) or break (-) reversibility. In particular, we show in Appendix B that the non-reversibility index of the process is related to the $\{\lambda_\ell^\pm\}$ coefficients through:

$$\zeta = \left(\frac{\sum(\lambda_\ell^-)^2}{\sum(\lambda_\ell^+)^2}\right)^{1/2}. \tag{8}$$

Thus, breaking reversibility requires the presence of skew-symmetric/odd terms. However, the decomposition does not immediately tell us how to *construct* such a non-reversible covariance function. Although one can show that the first term $A_1^+ f_1^+(\tau)$ must be positive definite, the addition of even a single $A_1^- f_1^-(\tau)$ odd term will not preserve positive definiteness in general, unless carefully specified. One of the main contributions of this work is to provide a constructive way of building sums of Kronecker products similar to Eq. 7, for which positive-definiteness is preserved while $\zeta$ can differ substantially from zero.

## 3.2 Planar non-reversible processes

To build intuition, we begin with a planar (two-output) process, $\boldsymbol{x}(t) = (x_1(t), x_2(t))^\top$, with zero mean and stationary matrix-valued covariance function $K(\cdot)$. If $x_1(t)$ and $x_2(t)$ are independent ($k_{ij}(\cdot) = \delta_{ij} f(\cdot)$) as in the original GPFA model (Yu et al., 2009), then the process is fully reversible. Consider, instead, the following construction:

$$K(\tau) = \underbrace{\begin{pmatrix} \sigma_1^2 & \sigma_1\sigma_2\rho \\ \sigma_1\sigma_2\rho & \sigma_2^2 \end{pmatrix}}_{A^+} f(\tau) + \alpha \underbrace{\begin{pmatrix} 0 & \sigma_1\sigma_2\sqrt{1-\rho^2} \\ -\sigma_1\sigma_2\sqrt{1-\rho^2} & 0 \end{pmatrix}}_{A^-} \mathcal{H}[f](\tau) \tag{9}$$

where $f(\cdot)$ is any scalar covariance function (an even function), $\mathcal{H}[f](\cdot)$ denotes its Hilbert transform (an odd function), $|\rho| \leq 1$ and $\alpha \in \mathbb{R}$. We show in Appendix C that Eq. 9 is a valid, positive semi-definite covariance, provided that $|\alpha| \leq 1$. Since $\mathcal{H}[f](0) = 0$, the first matrix on the r.h.s. parameterizes the instantaneous covariance $K(0)$ of the two processes (up to a positive scalar given by $f(0)$). Moreover, marginally, both $x_1(t)$ and $x_2(t)$ have temporal autocovariance function $f(\cdot)$.

Importantly $x_1$ and $x_2$ are now temporally correlated in such a way that reversibility is broken. In fact, Eq. 8 shows that $|\alpha|$ is related to the non-reversibility index $\zeta$ defined in Eq. 6 in the following

| stationary kernel $f(\tau)$ | Hilbert transform $\mathcal{H}[f](\tau)$ |
|---|---|
| $\exp(-\tau^2/2)$ | $2\pi^{-1/2}D(\tau/\sqrt{2})$ |
| $\cos(\omega_0\tau)$ | $\sin(\omega_0\tau)$ |
| $\sin(\omega_0\tau)/(\omega_0\tau)$ | $[1-\cos(\omega_0\tau)]/(\omega_0\tau)$ |
| $(1+\tau^2)^{-1}$ | $\tau(1+\tau^2)^{-1}$ |
| $\exp(-|\tau|)$ | $\left[e^{-\tau}\mathrm{Ei}(\tau)-e^{\tau}\mathrm{Ei}(-\tau)\right]/\pi$ |
| $\exp(-\tau^2/2)\cos(\omega_0\tau)$ | $\exp(-\tau^2/2)\sin(\omega_0\tau)+\exp(-\omega_0^2/2)\,\mathrm{Im}\,w((\tau+j\omega_0)/\sqrt{2})$ |

Table 1: **Hilbert transforms of usual scalar GP kernels.** Non-unit length-scales can be accommodated via a simple change of variable. Here, $D(\cdot)$ denotes the Dawson function, $\mathrm{Ei}(\cdot)$ is the exponential integral, and $w(\cdot)$ is the Faddeeva function.

way:

$$\zeta = |\alpha| \left[ \frac{2(1-\rho^2)}{(\sigma_1/\sigma_2)^2+(\sigma_2/\sigma_1)^2+2\rho^2} \right]^{1/2}, \tag{10}$$

which has a maximum of $|\alpha|$ when $\sigma_1 = \sigma_2$ and $\rho = 0$, i.e. for an instantaneously spherical process. Thus, Eq. 9 lets us construct planar GPs with arbitrary degrees of non-reversibility, with $\zeta$ ranging from 0 to 1.

For the construction of Eq. 9 to be of any use, one needs a practical way of evaluating the Hilbert transform of the marginal temporal covariance $f(\cdot)$. In Table 1, we provide a list of Hilbert transform pairs for several commonly used stationary GP kernels. Notably, we cover the case of the spectral mixture kernel (SM, last row of the table; Wilson and Adams, 2013), which currently achieves state-of-the-art results in GP-based extrapolation for one-dimensional timeseries. Although some of the Hilbert transforms that we were able to derive involve exotic functions, such as the Dawson and Faddeeva functions, these are readily available in most numerical programming environments. Moreover, they have analytical derivatives (Appendix D), such that they can easily be added to standard automatic differentiation software to enable automatic gradient computations for the model evidence e.g. in GP regression or GPFA (see below).

Fig. 1 illustrates the behavior of various spatially spherical planar GP kernels constructed from Eq. 9 with different kernels $f(\cdot)$. In cases where $\zeta = 1$, we emphasize that the time-reversed version of each of the samples shown (or indeed, of any subset thereof) has zero probability density under the prior from which it was drawn (Appendix C).

### 3.3 Fourier domain interpretation

To gain more insight into planar non-reversible GPs, we present an alternative construction of the process defined in Eq. 9, in the frequency domain. This construction can also serve as an alternative proof that Eq. 9 constitutes a valid GP covariance (see also Appendix C). We begin by noting that the Fourier transform of $\mathcal{H}[f]$ equals $-j\,\mathrm{sgn}(\omega)\widehat{f}(\omega)$, where $\widehat{f}(\omega)$ is the Fourier transform of $f$. In other words, $\mathcal{H}[f]$ is the real function that is phase shifted by $\pi/2$ away from $f$ at *all* frequencies. Thus, using the Wiener-Khinchin theorem, the Fourier-domain equivalent of Eq. 9 is:

$$\mathbb{E}\left[\widehat{x}_1(\omega)\overline{\widehat{x}_1(\omega)}\right] = \sigma_1^2\widehat{f}(\omega), \qquad \mathbb{E}\left[\widehat{x}_2(\omega)\overline{\widehat{x}_2(\omega)}\right] = \sigma_2^2\widehat{f}(\omega), \tag{11}$$

$$\mathbb{E}\left[\widehat{x}_1(\omega)\overline{\widehat{x}_2(\omega)}\right] = \sigma_1\sigma_2\widehat{f}(\omega)\left[\rho - \alpha\sqrt{1-\rho^2}\,j\,\mathrm{sgn}(\omega)\right], \tag{12}$$

where $\overline{\cdot}$ denote the complex conjugate and $\mathbb{E}[\cdot]$ denotes expectations w.r.t. the joint processes $(\widehat{x}_1, \widehat{x}_2)$ specified in the frequency domain. It is easy to verify that these (cross-)spectral densities can

be achieved by sampling the two processes according to:

$$\widehat{x}_1(\omega) = \sigma_1 \sqrt{\widehat{f}(\omega)} \left[ \widehat{\varepsilon}_1(\omega) \sqrt{1-\beta} + \widehat{\eta}(\omega) \sqrt{\beta} \right] \tag{13}$$

$$\widehat{x}_2(\omega) = \sigma_2 \sqrt{\widehat{f}(\omega)} \left[ \widehat{\varepsilon}_2(\omega) \sqrt{1-\beta} + \widehat{\eta}(\omega) \exp(j \operatorname{sgn}(\omega)\varphi) \sqrt{\beta} \right] \tag{14}$$

where $\varepsilon_1(t)$, $\varepsilon_2(t)$ and $\eta(t)$ are independent white noise processes with unit variance, and $\beta$ and $\varphi$ obey the following parameter correspondance: $\rho = \beta \cos(\varphi)$ and $\alpha \sqrt{1-\rho^2} = \beta \sin(\varphi)$. In other words, $x_1(t)$ and $x_2(t)$ are each entrained to a common latent process $\eta(t)$ with some degree of coherence $\beta$, and some frequency-independent phase lag (0 for $x_1$ without loss of generality, and $\varphi$ for $x_2$). In particular, for an instantaneously uncorrelated joint process ($\rho = 0$), we have $\varphi = \pi/2$ and $\beta = \alpha$. Spatial correlations $\rho \neq 0$ can be introduced through phase shifts $\varphi$ different from $\pi/2$.

### 3.4   Higher-dimensional non-reversible priors

The non-reversible planar processes described in Section 3.2 can be extended to $M$-output processes with $M > 2$ in several ways. Here, we focus on simple combinations of individual planes, but see Appendix E for potentially more flexible approaches. Specifically, we construct an M-output covariance function as a superposition of planar processes of the form of Eq. 9:

$$K(\tau) = \sum_{1 \leq i < j \leq M} A^{ij+} f_{ij}(\tau) + \alpha_{ij} A^{ij-} \mathcal{H}[f_{ij}](\tau) \tag{15}$$

with

$$A_{uv}^{ij+} = \sigma_{ij,1}^2 \delta_{ui}\delta_{vi} + \sigma_{ij,2}^2 \delta_{uj}\delta_{vj} + \sigma_{ij,1}\sigma_{ij,2}\rho_{ij}(\delta_{ui}\delta_{vj} + \delta_{uj}\delta_{vi}) \quad \text{(symm. PSD matrix)} \tag{16}$$

$$A_{uv}^{ij-} = \sigma_{ij,1}\sigma_{ij,2}\sqrt{1-\rho_{ij}^2}\,(\delta_{ui}\delta_{vj} - \delta_{uj}\delta_{vi}) \quad\quad\quad\quad \text{(skew-symm. matrix)} \tag{17}$$

and $|\alpha_{ij}| \leq 1$. Note that $A_{ij}^+$ and $A_{ij}^-$ are defined in the same way as $A^+$ and $A^-$ in Eq. 9, and involve only two of the latent dimensions ($i$ and $j$). Thus, each term in the sum describes a covariance over a pair of dimensions. This sum of planar kernels is motivated by the general decomposition in Eq. 7, though it does not obey the orthogonality constraints therein. In our GPFADS experiments, we further truncate this sum to $M/2$ non-overlapping planes with no shared latent dimensions.

## 4   Experiments

In this section, we begin by demonstrating the utility of non-reversible GP priors for modeling time-series data produced by dynamical systems. We then go on to introduce such non-reversible priors in GPFA, and show that GPFADS recovers the Markov state of low-dimensional dynamical systems embedded in high-dimensional data. We also apply GPFADS to primary motor cortex data, where it automatically discovers rotational dynamics that have been shown to emerge during reaching movements (Churchland et al., 2012).

### 4.1   Non-reversible GP priors better capture dynamics

Fig. 2 illustrates the relevance of non-reversible planar processes of the type of Eq. 9 for modelling multivariate time-series produced by dynamical systems. We simulated state trajectories of the classical pendulum ($\dot{x}_1 = x_2$ and $\dot{x}_2 = -\sin(x_1)$) as well as the Duffing oscillator ($\dot{x}_1 = x_2$ and $\dot{x}_2 = x_1 - x_1^3$), starting from random initial conditions (Fig. 2A). We then fitted a GP with kernel given by Eq. 9, either with $\alpha$ optimized as part of the fit ('non-rev'), or pinned to zero ('rev'). The non-reversible model consistently outperformed the reversible one on cross-validated marginal likelihood (Fig. 2B). Importantly, by optimizing the non-reversibility parameter $\alpha$, the model learned to capture the phase relationship between $x_1$ and $x_2$, resulting in much better extrapolations than for the reversible model (Fig. 2C). In particular, it was possible to accurately reconstruct $x_2$ by only conditioning on $x_1$ and the initial condition for $x_2$.

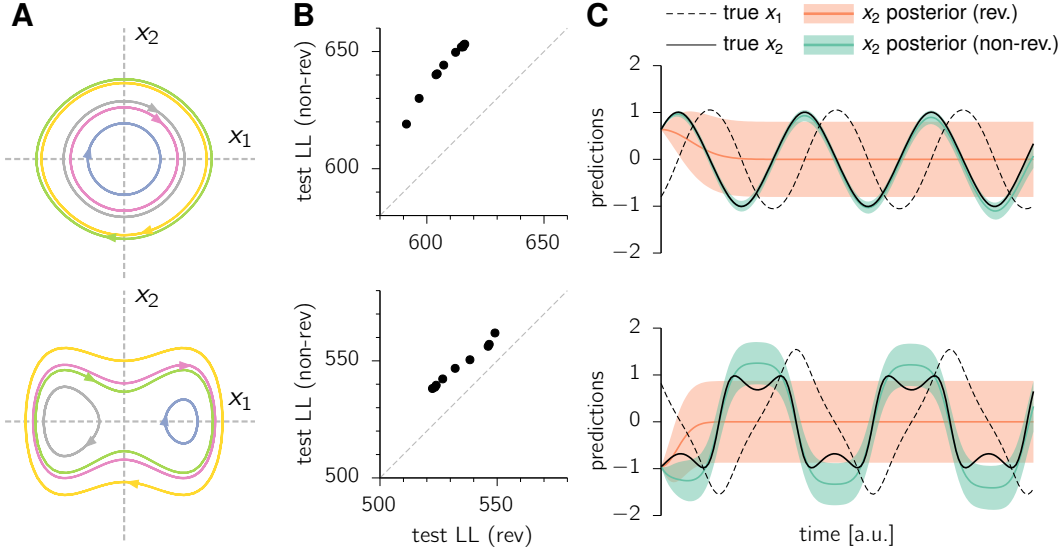

Figure 2: **Non-reversible GP regression on state trajectories of the classical pendulum (top) and the Duffing oscillator (bottom). (A)** Five of the 20 planar trajectories $(x_1(t), x_2(t))$ used for training. Each bout of training data (color-coded) is of the same duration and contains several cycles, the exact number of cycles depending on the (conserved) Hamiltonian energy. **(B)** Marginal likelihood for 10 individual test trajectories, for the planar reversible SE kernel (x-axis, $\alpha$ pinned to zero) and its non-reversible counterpart (y-axis, $\alpha$ optimized to $0.99$ for the pendulum, and $0.9$ for the Duffing oscillator). **(C)** Posterior over $x_2(t)$ in each model, conditioned on the full time course of $x_1$ (dashed black) but only on the first time bin of $x_2$. Ground truth $x_2(t)$ is in solid black.

## 4.2 GPFADS: recovering embedded latent dynamical systems

We now combine GPFA with the non-reversible multi-output priors introduced in Section 3, and call this combination GPFADS, Gaussian Process Factor Analysis with Dynamical Structure. In this section, we investigate the extent to which this extension of GPFA allows us to learn something about the dynamics that might underlie a set of multivariate time-series. We reason that noise tends to be more time-reversible than signal generated from a dynamical process, such that placing a non-reversible prior over latent trajectories might let us demix signal with dynamical structure from noise. To demonstrate this, we embedded a 2D dynamical system, the Van der Pol oscillator ($\dot{x}_1 = x_2$ and $\dot{x}_2 = (1 - x_1^2)x_2 - x_1$), into a higher dimensional ambient space ($N = 6$). We also included another (orthogonal) latent plane in which activity was drawn independently for each of the two dimensions from a GP with squared-exponential kernel. We matched the timescales of this reversible "distractor" process to the characteristic timescales of the Van der Pol oscillator.

We trained both GPFADS and GPFA with $M = 4$ latent dimensions on the same set of 50 trajectories, with the Van der Pol oscillator seeded with random initial states in each one. For GPFADS, we used the kernel described in Eq. 15 with all $f_{ij}(\cdot)$ set to the squared-exponential kernel (with independent hyperparameters), and with the sum over $(i, j)$ planes restricted to $(1, 2)$ and $(3, 4)$ – i.e. two independent, orthogonal planes. For GPFA, we placed independent squared-exponential priors on each of the 4 latent dimensions (Yu et al., 2009). We note that the two models had the same number of parameters: GPFA had two more timescales than GPFADS, but the latter model had two learnable non-reversibility parameters $\alpha_{12}$ and $\alpha_{34}$.

We found that GPFADS successfully demixed the plane containing the oscillator from that containing the distractor process. Indeed, after training, one of the two latent planes was highly non-reversible ($|\alpha_{12}| = 0.88$, vs. $|\alpha_{34}| = 0.13$), and posterior trajectories in the non-reversible plane recovered the state trajectories of the Van der Pol oscillator (Fig. 3, left). In contrast, despite GPFA being able to correctly learn the various timescales in the latent processes, it failed to demix signal from noise, such that no clear dynamical picture emerged (Fig. 3, right). GPFA also performed worse than GPFADS based on the cross-validated marginal likelihood (not shown).

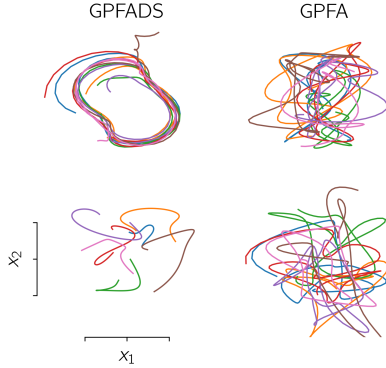

GPFADS      GPFA

Figure 3: **GPFADS recovers the state trajectories of a dynamical system embedded in a high-dimensional ambient space**. GPFADS and GPFA latent trajectories (posterior mean) inferred from observations arising as a mixture of trajectories produced by the Van der Pol oscillator and a noisy process of equal variance, each embedded in 6D and added together with white noise (see text for details). For GPFADS, we show the two latent planes over which separate non-reversible planar priors were placed. For GPFA, we only show two arbitrarily chosen planes, but any other combination of latent dimensions resulted in similar unstructured trajectories.

## 4.3 Uncovering rotational dynamics in M1

Collective neural activity in monkey and human primary motor cortex (M1) has been shown to embed strong rotational latent dynamics (Churchland et al., 2012; Pandarinath et al., 2018). The extraction of these dynamics has historically relied on a method called jPCA purposely designed to extract latent rotations wherever they exist. At the heart of such analysis is the desire to reveal dynamical structure in population activity, but one potential concern with jPCA is that it biases this search towards pure rotations, whereas dynamics could in fact be of other forms. With this in mind, we applied GPFADS to M1 population recordings performed in monkey during reaching (Fig. 4; Churchland et al., 2012), to investigate the extent to which rotational dynamics are revealed by a method which does not explicitly look for them, but only indirectly through a search for non-reversible behavior.

The data consisted of $N = 218$ neurons whose activity time-course was measured in each of 108 different movement conditions, and aligned temporally to the onset of movement. For each neuron, activity was averaged over hundreds of stereotypical repetitions of each movement, and further smoothed and 'soft-normalized' (Churchland et al., 2012). We fitted GPFADS with an increasing number of latent dimensions ($M = 2, 4, 6$; Fig. 4A-C). For each $M$, we used a non-reversible $M$-output GP kernel of the form described in Eq. 15, though we restricted the number of possible planes to $M/2$ independent planes with no shared dimensions. As $C$ was not constrained to be orthogonal we fixed $\rho = 0$, as any prior spatial correlations in a given plane could in this case be absorbed by a rotation of the corresponding two columns of $C$. Due to the smoothing of neural activity at pre-processing stage (which we did not control), we found that fitting GPFA(DS) was prone to so-called Heywood cases where some diagonal elements of $R$ in Eq. 1 converge to very small values if allowed to (Heywood, 1931; Martin and McDonald, 1975). To circumvent this, here we constrained $R \propto I$, but note that this issue would likely not arise in the analysis of single-trial, spiking data.

For $M = 2$, GPFADS learned a mixing matrix $C$ that was near identical (up to a rotation) to the one learned by the original GPFA model (not shown). This is not surprising: for $M = 2$, a good fit for GPFADS and GPFA alike is likely to be one in which the two columns of $C$ capture the most data variance, regardless of how non-reversible latent activity happens to be in the plane defined by these two columns. Nevertheless, we found that GPFADS learned a non-reversibility parameter $\alpha_{12} = 0.72$ in this case, indicating that dynamics were fairly non-reversible in this top subspace. Importantly, when GPFADS was fit with a larger latent dimension ($M = 4$ or 6; Fig. 4B-C), it cleanly segregated latent trajectories into strongly rotatory planes ($|\alpha_{ij}| \in \{0.94, 0.85, 0.95\}$) and planes that absorbed remaining fluctuations with less apparent dynamical structure ($|\alpha_{ij}| \in \{0.35, 0.59\}$). With increasing latent dimension, we found that allowing for non-reversibility in the prior yielded increasing benefits over an equivalent model where all $\{\alpha_{ij}\}$ parameters were set to zero (and the other parameters optimized as normal; Fig. 4D).

## 5 Discussion

A great challenge in neuroscience is to unravel the dynamical mechanisms that underlie neuronal computations. As a first step to this, many data analysis methods focus on inferring latent processes which compactly summarize observations of neural activity in various tasks. Gaussian process-based methods, such as GPFA (Yu et al., 2009), offer data-efficient ways of extracting such latent

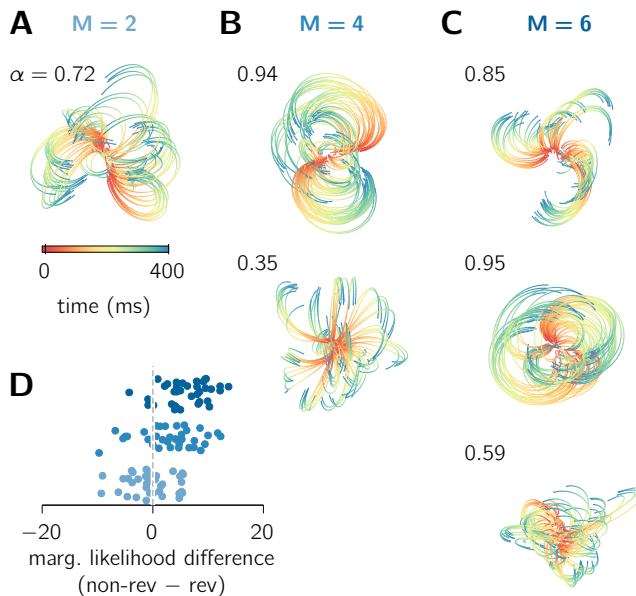

**A** M = 2  **B** M = 4  **C** M = 6

$\alpha = 0.72$   0.94   0.85

0     400
time (ms)

0.35   0.95

0.59

**D**

−20     0     20
marg. likelihood difference
(non-rev − rev)

Figure 4: **Discovering latent dynamical structure in primary motor cortex with GPFADS. (A-C)** GPFADS latent trajectories (posterior mean) for $M = 2$ (A, one plane), $M = 4$ (B, two planes) and $M = 6$ (C, three planes), for all movement conditions. The value of $|\alpha_{ij}|$ in Eq. 15 for each of the $M/2$ planes is shown near each plot. All plots share the same scale, and planes are ordered from top to bottom by decreasing total variance in the learnt prior ($\sigma_i^2 + \sigma_j^2$). **(D)** Difference in marginal likelihood for each of 35 trajectories in the test set, between GPFADS (in which the non-reversibility parameters $\{\alpha_{ij}\}$ are learned) and a similar model where these parameters are pinned to zero. Colors refer to the 3 models (see title colors in A-C). Results were found to be highly consistent over independent splits of the 108 conditions into train and test sets.

processes along with associated uncertainty. However, these methods fail to explicitly capture the dynamical nature of neural activity beyond basic smoothness properties. Here, we set out to impart some notion of "dynamics" to GP-based models, using temporal non-reversibility as a proxy for dynamics. We introduced a measure of second-order non-reversibility and derived a new family of GPs for which any sample has lower probability of occurring in reverse. We found that these priors outperform standard reversible ones on a number of datasets known to emanate from dynamical systems, including recordings from primary motor cortex.

An instance of a non-reversible kernel was introduced previously by Parra and Tobar (2017), which extended the spectral mixture model to a multi-output covariance expressing a variety of time delays and phase lags between dimensions (see also Appendix G). Here we have taken a different approach using the Hilbert transform, which – unlike Parra and Tobar (2017)'s kernel – admits a Kronecker factorization enabling scalability to large datasets.

While temporal non-reversibility is an expected property of most dynamical systems in which state trajectories follow a lawful flow-field (unless they are strongly input driven), it is an incomplete characterization. In particular, the trajectories generated by our non-reversible GP models (Fig. 1) often cross over, which would not occur in an autonomous dynamical system where the flow would be entirely determined by the momentary state (unless the state is only partially observed). It would be interesting to explore non-reversible GP kernels that also express this complementary property — while the cosine kernel in Table 1 satisfies both properties, it is unclear if a more general, less constraining form exists. Such models might be particularly well-suited for modeling population activity in M1 which is markedly "untangled" (Russo et al., 2018), i.e. lacks cross-overs.

Other non-probabilistic methods have been proposed for reducing the dimensionality of datasets whilst preserving "dynamical" characteristics. For example, Dynamical Components Analysis (DCA; Clark et al., 2019) seeks the lower-dimensional subspace that maximizes predictive information. The authors showed that DCA can successfully recover the Markov state of low-dimensional dynamical systems embedded in high dimensions, similarly to GPFADS in Fig. 3. GPFADS is even more closely related to another dimensionality reduction technique which we have proposed previously, Sequential Components Analysis (SCA; Rutten et al., 2020), in which the low-dimensional projection is chosen to maximize our measure of second-order non-reversibility in Eq. 6. However, in contrast to SCA, GPFADS does not explicitly seek to maximize this measure, but instead automatically learns the degree of non-reversibility (determined by the kernel hyper-parameters) that best explains the data.

## Acknowledgments

We thank Ta-Chu Kao for sharing his Cholesky-GPFA derivation, and Richard Turner for discussions.

## Broader impact

From molecules to stock-markets, from short timescales to long, the arrow of time can be seen in the evolution of natural living systems. Indeed, the dynamics of natural living systems are not time-reversible, they depart from "thermodynamic equilibrium" (Gnesotto et al., 2018). Despite the ubiquity and importance of non-reversibility, there is a paucity of methods for exploring the spatio-temporal structure of irreversibility in multivariate time series. The new class of non-reversible covariance functions we developed makes use of this intrinsic property, offering the potential of exploiting this natural feature in a range of data analysis algorithms.

In general, quantifying and tracking changes in reversibility over time could be useful in detecting the early onset of real world events. More specifically, we expect that one of the larger impacts of the method will be in the field of brain-machine interfaces (BMI) and neuroprosthetics. BMIs have the possibility of revolutionizing how we live. Optimizing the interface both at the hardware and software level is key to making this a reality. Regarding software, identifying actionable latent variables embedded in high-dimensional neural activity is of particular importance in facilitating communication. Given that behavior is non-reversible, the neural activity that causally drives this behavior is likely to also be non-reversible, thus seeking latents with such property seems highly promising. Moreover, BMI algorithms often need to be run online which the scalability of our method would also permit. These applications come with ethical and societal concerns, in particular regarding privacy and responsibility. These ethics challenges are being actively investigated by the field of bioethics (Clausen, 2008) and the broader community; and we hope that such considerations will continue to shape the future research and reality.

## Funding disclosure

This work was funded by the Gatsby Charitable Foundation (VMSR, MS), the Simons Foundation (SCGB 543039; MS) and a Janelia HHMI Graduate Fellow Research Scholarship (VMSR). Competing interests: none to declare.

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
