[Supplementary Material]

# Appendix — Non-reversible Gaussian processes for identifying latent dynamical structure in neural data

## A  Boundedness of the reversibility index

In this section, we prove that for a stationary multi-output GP, the reversibility index defined in Eq. 6 is bounded between 0 and 1. Recall the definition of the (squared) reversibility index:

$$\zeta^2 = \frac{\int_{-\infty}^{\infty} \|K(\tau) - K(-\tau)\|_{\mathrm{F}}^2 \, d\tau}{\int_{-\infty}^{\infty} \|K(\tau) + K(-\tau)\|_{\mathrm{F}}^2 \, d\tau}. \tag{18}$$

Since $\|X\|_{\mathrm{F}}^2 = \mathrm{Tr}(XX^\top)$, we can expand this into:

$$\zeta^2 = \frac{\mathrm{Tr} \int_{-\infty}^{\infty} K(\tau)K(\tau)^\top \, d\tau - \mathrm{Tr} \int_{-\infty}^{\infty} K(\tau)K(\tau) \, d\tau}{\mathrm{Tr} \int_{-\infty}^{\infty} K(\tau)K(\tau)^\top \, d\tau + \mathrm{Tr} \int_{-\infty}^{\infty} K(\tau)K(\tau) \, d\tau}. \tag{19}$$

The first term in both the numerator and the denominator is non-negative because the integral of outer products $K(\tau)K(\tau)^\top$ is positive semi definite (PSD). We are left to show that the second term is non-negative too, which would then imply $0 \le \zeta \le 1$. From Parseval's theorem, we have that:

$$\int_{-\infty}^{\infty} K(\tau)K(\tau) \, d\tau = \int_{-\infty}^{\infty} \widehat{K}(\omega)\overline{\widehat{K}(\omega)} \, d\omega \tag{20}$$

where $\widehat{\cdot}$ denotes the Fourier transform (assumed to exist) and $\overline{\cdot}$ denotes the complex conjugate. Furthermore, from Cramér's theorem, since $K(\cdot)$ is a stationary cross-covariance function, $\widehat{K}(\omega)$ is Hermitian PSD, and so is $\overline{\widehat{K}(\omega)}$, for any $\omega$. Finally, for any two Hermitian PSD matrices, A and B, it can be shown that $\mathrm{Tr}(AB) \ge 0$ (Theorem 4.3.53 in Horn and Johnson, 2012). Thus, we have shown that:

$$\mathrm{Tr} \int_{-\infty}^{\infty} K(\tau)K(\tau)^\top d\tau \ge 0 \tag{21}$$

$$\mathrm{Tr} \int_{-\infty}^{\infty} K(\tau)K(\tau)d\tau \ge 0 \tag{22}$$

from which we can conclude that: $0 \le \zeta \le 1$.

## B  General Kronecker decomposition of stationary multi-output covariances

Here, we prove the existence of the decomposition of Eq. 7. Let $K(\tau)$ be a matrix-valued (i.e. multi-output) cross-covariance function; we assume that each scalar cross-covariance function $K_{ij}(\cdot)$ is in $L^2$. From the singular value decomposition theorem for compact operators in Hilbert spaces (Crane et al., 2020), there exist a basis set of $M^2$ functions $\{f_\ell(\cdot) \in L^2\}_{\ell=1}^{M^2}$, a matching set of matrices $\{A_\ell \in \mathbb{R}^{M \times M}\}_{\ell=1}^{M^2}$, and a corresponding set of positive singular values $\{\lambda_\ell \in \mathbb{R}^+\}_{\ell=1}^{M^2}$ such that:

$$K(\tau) = \sum_{\ell=1}^{M^2} \lambda_\ell A_\ell \cdot f_\ell(\tau) \tag{23}$$

with the following orthonormality conditions:

$$\mathrm{Tr}\left(A_\ell A_{\ell'}^\top\right) = \delta_{\ell\ell'} \tag{24}$$

$$\int f_\ell(\tau) f_{\ell'}(\tau) \, d\tau = \delta_{\ell\ell'}. \tag{25}$$

We will refer to Eq. 23 as a "generalized SVD". We now show that the pairs $\{A_\ell, f_\ell\}$ are either symmetric/even, or skew-symmetric/odd, as stated in Section 3.1. Let $\tilde{\boldsymbol{k}}(\tau) = \mathrm{vec}(K(\tau)) \in \mathbb{R}^{M^2}$ be the vectorized cross-covariance matrix at time lag $\tau$. Let $P \in \{0,1\}^{M^2 \times M^2}$ be the commutation

matrix operating on the vectorized space of all $M^2$ pairs of outputs (i.e. the space of $\tilde{\boldsymbol{k}}$), which swaps index $(i + jM)$ with index $(j + iM)$. In other words, for any matrix $X \in \mathbb{R}^{M \times M}$, we have that $P \operatorname{vec}(X) = \operatorname{vec}(X^\top)$. Let $\mathcal{R}$ be the reflection (or 'time-reversal') operator in $L^2$, such that $\mathcal{R}[f](\tau) = f(-\tau)$. It is easy to show that any multi-output cross-covariance function $K(\tau)$ obeys the symmetry $K(-\tau) = K(\tau)^\top$, which can also be written as:

$$\mathcal{R}[\tilde{\boldsymbol{k}}](\tau) = P\tilde{\boldsymbol{k}}(\tau) \tag{26}$$

where $\mathcal{R}[\cdot]$ is applied elementwise to the element functions in $\tilde{\boldsymbol{k}}$. In other words, reversing time and transposing space are two equivalent operations. The generalized SVD in Eq. 23 can be written in vectorized form as

$$\tilde{\boldsymbol{k}}(\tau) = \underbrace{(\tilde{\boldsymbol{a}}_1, \ldots, \tilde{\boldsymbol{a}}_{M^2})}_{U} \underbrace{\operatorname{diag}(\lambda_1, \ldots, \lambda_{M^2})}_{S} \underbrace{(f_1(\tau), \ldots, f_{M^2}(\tau))^\top}_{\boldsymbol{v}(\tau)} \tag{27}$$

where $\tilde{\boldsymbol{a}}_\ell = \operatorname{vec}(A_\ell)$. Thus, the symmetry of Eq. 26 can be re-expressed as

$$(PU)S(\mathcal{R}[\tilde{\boldsymbol{v}}](\tau)) = US\tilde{\boldsymbol{v}}(\tau). \tag{28}$$

Since the two permutation operators $P$ and $\mathcal{R}$ preserve orthonormality (of matrices and square-integrable functions, respectively), the l.h.s. of Eq. 28 is a valid SVD for $K(\tau)$. Furthermore, when the singular values are distinct and kept in decreasing order, the generalized SVD of $\tilde{\boldsymbol{k}}(\tau)$ is unique up to a *simultaneous* change of sign in any pair $(\tilde{\boldsymbol{a}}_\ell, f_\ell(\cdot))$. Therefore, we must have that

$$PU = U\pm \tag{29}$$
$$\mathcal{R}[\tilde{\boldsymbol{v}}](\tau) = \pm\tilde{\boldsymbol{v}}(\tau) \tag{30}$$

where $\pm$ is a (shared) diagonal matrix composed of $+1$ and $-1$ elements only. Importantly, the matrix $\pm$ is the same in the two equations. From the spatial transpose meaning of $P$ and time-reversal meaning of $\mathcal{R}$, Eqs. 29 and 30 therefore imply that for any $\ell$:

- if $\pm_\ell = +1$, then $\tilde{\boldsymbol{a}}_\ell$ is invariant to the transposition operator $P$ and $f_\ell(\cdot)$ is invariant to time reversal, implying that $A_\ell$ is a symmetric matrix and $f_\ell$ is an even function,

- if $\pm_\ell = -1$, then both $\tilde{\boldsymbol{a}}_\ell$ and $f_\ell$ have their sign flipped by spatial transposition and time reversal, respectively, implying that $A_\ell$ is a skew-symmetric matrix and $f_\ell$ is an odd function.

The decomposition of Eq. 7 follows from a simple, now justified renaming of symmetric/even and skew-symmetric/odd pairs as $\{A_\ell^\pm, f_\ell^\pm\}$. Moreover, we now see that the $+$ (resp. $-$) terms in Eq. 27 correspond to the generalized SVD of the symmetric (resp. antisymmetric) part of the covariance function, $K(\tau) + K(-\tau)$ (resp. $K(\tau) - K(-\tau)$) in Eq. 18. Given the known relationship between the squared Frobenius norm of a linear operator and the sum of its squared singular values, this shows that the non-reversibility index $\zeta$ can also be calculated using Eq. 8.

## C Construction of valid non-reversible planar kernels

Here, we prove that Eq. 9 is a valid 2-output covariance function. We focus the proof on the purely spherical case, $\sigma_1 = \sigma_2 = 1$ and $\rho = 0$ – extension to arbitrary instantaneous covariances is straightforward but notationally cumbersome (and a more general proof was in fact already given in Section 3.3).

In the frequency domain, the Hilbert transform has a simple interpretation as a constant phase shift of $\pi/2$ at all frequencies; specifically, $\widehat{\mathcal{H}[f]}(\omega) = -j \cdot \operatorname{sgn}(\omega)\widehat{f}(\omega)$, where $\widehat{\phantom{x}}$ denotes the Fourier transform and $j^2 = -1$. As any scalar covariance function, $f$ is even, and therefore $\widehat{f}$ is real and even. In contrast, $\widehat{\mathcal{H}[f]}$ is imaginary and odd. We now derive the eigendecomposition of $K(\cdot)$, and show that all eigenvalues are positive. Due to the hybrid nature of $K(\cdot)$ (discrete space, continuous time), an eigenfunction of $K(\cdot)$ is a time-varying 2-dimensional "vector" $[g_1(t), g_2(t)]^\top$ which satisfies

$$\int_{-\infty}^{\infty} K(\tau - t) \begin{bmatrix} g_1(\tau) \\ g_2(\tau) \end{bmatrix} d\tau = \lambda \begin{bmatrix} g_1(t) \\ g_2(t) \end{bmatrix} \tag{31}$$

or equivalently,

$$\int_{-\infty}^{\infty} \left[ \begin{array}{c} f(\tau)g_1(\tau + t) + \alpha \mathcal{H}[f](\tau)g_2(\tau + t) \\ f(\tau)g_2(\tau + t) - \alpha \mathcal{H}[f](\tau)g_1(\tau + t) \end{array} \right] d\tau = \lambda \left[ \begin{array}{c} g_1(t) \\ g_2(t) \end{array} \right]. \tag{32}$$

Knowing that the eigenfunctions of any stationary scalar kernel are the Fourier modes, $e^{j\omega t}$, we make the following ansatz for the eigenfunctions of $K(\cdot)$:

$$\boldsymbol{g}_\omega^{\pm}(t) = \left[ \begin{array}{c} e^{j\omega t} \\ b^{\pm} e^{\pm j\omega t} \end{array} \right] \qquad \text{for some } b^{\pm} \in \mathbb{C}, \tag{33}$$

(with the understanding that the two $\pm$ signs are "tied", i.e. they are either both $+$ or both $-$). Next, we note that for any scalar function $h$ and $\omega \in \mathbb{R}$,

$$\int_{-\infty}^{\infty} h(\tau)e^{\pm j\omega(t+\tau)}d\tau = e^{\pm j\omega t} \int_{-\infty}^{\infty} h(\tau)e^{\pm j\omega\tau}d\tau \tag{34}$$

$$= e^{\pm j\omega t} \int_{-\infty}^{\infty} h(\tau)e^{-j(\mp\omega)\tau}d\tau \tag{35}$$

$$= e^{\pm j\omega t}\widehat{h}(\mp\omega) \tag{36}$$

where the last equality follows from the definition of the Fourier transform $\widehat{h}(\cdot)$. Thus, for $\boldsymbol{g}_\omega^{\pm}(t)$ to be an eigenfunction of $K(\cdot)$ with eigenvalue $\lambda_\omega^{\pm}$, $b^{\pm}$ must satisfy:

$$\widehat{f}(\omega) + b^{\pm}\alpha\widehat{\mathcal{H}[f]}(\mp\omega) = \lambda_\omega^{\pm} \tag{37}$$

$$-\alpha\widehat{\mathcal{H}[f]}(\mp\omega) + b^{\pm}\widehat{f}(\omega) = b^{\pm}\lambda_\omega^{\pm} \tag{38}$$

(easily obtained from Eq. 32 and some straightforward algebra). Inserting $\widehat{\mathcal{H}[f]}(\omega) = -j \cdot \text{sgn}(\omega)\widehat{f}(\omega)$, the above system of equations imply:

$$\begin{aligned} \text{either} \quad b^{\pm} &= +j \quad \rightarrow \quad \lambda_\omega^{\pm} = (1 - \text{sgn}(\pm\omega)\alpha)\,\widehat{f}(\omega) \\ \text{or} \quad b^{\pm} &= -j \quad \rightarrow \quad \lambda_\omega^{\pm} = (1 + \text{sgn}(\pm\omega)\alpha)\widehat{f}(\omega). \end{aligned} \tag{39}$$

Since $f$ is itself a valid kernel whose real eigenvalues $\widehat{f}(\omega)$ are all strictly positive, we conclude that $K(\cdot)$ is a valid 2-output covariance function (i.e. all $\lambda_\omega^{\pm} \geq 0$) if, and only if, $|\alpha| \leq 1$.

This derivation also provides a key connection between our measure of second-order non-reversibility, $\zeta$, and one's intuitive understanding that a maximally non-reversible process should never "turn back on itself". Specifically, we note that when $|\alpha| = 1$, Eq. 39 implies that half of the eigenvalues of $K(\cdot)$ are exactly zero. By inspection of the corresponding eigenfunctions in Eq. 32, we see that for any sample trajectory $(x_1(t), x_2(t))^\top$ from $K(\cdot)$, which must lie in the span of the eigenfunctions with non-zero eigenvalues, the time-reversed trajectory $(x_1(-t), x_2(-t))$ evolves in the span of the eigenfunctions with zero eigenvalues. Therefore, these time-reversed trajectories have zero probability density under our non-reversible prior when $|\alpha| = 1$; this extreme case corresponds to $\zeta = 1$, i.e. maximal non-reversibility (c.f. Eq. 10).

## D   Derivatives of special functions

Optimization of the hyperparameters requires evaluating the gradient of the marginal likelihood, which in turn requires gradients of all the functions involved in the multi-output kernel. We give details of the gradients for the functions listed in Table 1 and present them graphically in Fig. 5. These gradients are simple to evaluate numerically, enabling the addition of our non-reversible kernels to standard automatic differentiation systems.

**Dawson function**   The Hilbert transform of the squared-exponential kernel involves the so-called Dawson function, given below along with its derivative:

$$D(\tau) \triangleq e^{-\tau^2} \int_0^\tau e^{s^2}\, ds \qquad\qquad \frac{dD}{d\tau} = 1 - 2\tau D(\tau). \tag{40}$$

Figure 5: **Commonly used GP covariance functions and their Hilbert transforms**. Please refer to Table 1 for details.

**Exponential integral**   The Hilbert transform of the exponential kernel involves the exponential integral function:

$$\text{Ei}(\tau) \triangleq \int_{-\infty}^{\tau} \frac{e^s}{s}\,ds \qquad\qquad \frac{d\text{Ei}}{d\tau} = \frac{e^\tau}{\tau}. \tag{41}$$

**Faddeeva function**   The Hilbert transform of the spectral mixture kernel (Wilson and Adams, 2013) involves the imaginary component of the Faddeeva function $w(\cdot)$ (related to the complex error function; Zaghloul and Ali, 2012). This is a new result that we derived which we were not able to find in the existing literature. The Faddeeva function is available in most numerical computing environments, and can be expressed as:

$$w(z) = V(x,y) + jL(x,y) \qquad \text{with } z = x + jy \tag{42}$$

where $V(x,y)$ and $L(x,y)$ are the real and imaginary Voigt functions respectively. Gradients are given by:

$$\frac{\partial L(x,y)}{\partial x} \quad = \quad -\frac{\partial V(x,y)}{\partial y} \quad = \quad -2\,\text{Im}[zw(z)] + \frac{2}{\sqrt{\pi}} \tag{43}$$

$$\text{and} \quad \frac{\partial L(x,y)}{\partial y} \quad = \quad \frac{\partial V(x,y)}{\partial x} \quad = \quad -2\,\text{Re}[zw(z)]. \tag{44}$$

Evaluating these expressions only requires evaluating the Fadeeva function itself.

# E   Beyond non-reversible planar processes: construction of higher-dimensional non-reversible covariance functions

In Section 3.4, we built non-reversible M-output GP covariances as superpositions of planar kernels, each associated with a (potentially) different scalar covariance function. Here, we introduce an alternative construction motivated by the following considerations of model degeneracies. In Eq. 15, if (i) the marginal variances in each plane are all identical ($\sigma_{ij,1} = \sigma_{ij,2} = \sigma$), (ii) each planar process is spherical ($\rho_{ij} = 0$) and (iii) the scalar covariance functions $f_{ij}(\tau)$ are all the same $f$, then $K(\tau)$ is over-parametrized. Indeed, it can then be rewritten as

$$K(\tau) \propto \underbrace{I_M}_{A^+} f(\tau) + \underbrace{\begin{pmatrix} 0 & \alpha_{12} & \alpha_{13} & \cdots \\ -\alpha_{12} & 0 & \alpha_{23} & \cdots \\ -\alpha_{13} & -\alpha_{23} & 0 & \cdots \\ \vdots & \vdots & \vdots & \ddots \end{pmatrix}}_{A^-} \mathcal{H}[f](\tau) \tag{45}$$

Now, since $A^-$ is antisymmetric, there exists a unitary transformation of the latent space in which this covariance becomes

$$K(\tau) \propto I_M f(\tau) + \begin{pmatrix} 0 & \omega_1 & 0 & 0 & \cdots \\ -\omega_1 & 0 & 0 & 0 & \cdots \\ 0 & 0 & 0 & \omega_2 & \cdots \\ 0 & 0 & -\omega_2 & 0 & \cdots \\ \vdots & \vdots & \vdots & & \ddots \end{pmatrix} \mathcal{H}[f](\tau) \tag{46}$$

where $\{\pm j\omega_1, \pm j\omega_2, \ldots\}$ are the imaginary conjugate eigenvalues of $A^-$. In GPFADS, this unitary transformation can be absorbed in the mixing matrix $C$ (Eq. 1). Thus, in this configuration, our model does not truly possess $M(M-1)/2$ free parameters as the parametrization of Eq. 15 suggests, but only $M/2$, as Eq. 46 reveals. In other words, one can always rotate the latent space and directly parametrize a set of independent planes – in which case one must enforce $|\omega_i| < 1\}$ to ensure positive definiteness.

For these reasons, we also propose this alternative construction:

$$K(\tau) = \sum_{q=1}^{Q} U_q \left( \begin{bmatrix} A_{q1}^+ & 0 & \ddots \\ 0 & A_{q2}^+ & 0 \\ \ddots & 0 & \ddots \end{bmatrix} f_q(\tau) + \begin{bmatrix} \alpha_{q1} A_{q1}^- & 0 & \ddots \\ 0 & \alpha_{q2} A_{q2}^- & 0 \\ \ddots & 0 & \ddots \end{bmatrix} \mathcal{H}[f_q](\tau) \right) U_q^\top \quad (47)$$

with some scalar covariance functions $\{f_q(\cdot)\}$, unitary matrices $\{U_q\}$, non-reversibility parameters $|\alpha_{qi}| < 1$ associated with each of the $M/2$ planes, and $2 \times 2$ matrix blocks $\{A_{qi}^+, A_{qi}^-\}$. The latter are parameterized exactly as the symmetric and antisymmetric matrices $A^+$ and $A^-$ in Eq. 9. We note that when such a kernel is used in GPFA, $U_1$ can be set to the identity matrix without loss of generality since it corresponds to a rotation of the latent space that can be absorbed in the mixing matrix $C$ (Eq. 1).

# F   Implementation notes and scalability

## F.1   Stable computation of the log marginal likelihood

The log marginal likelihood in GPFA(DS) can be computed in a stable way as follows. Recall its expression:

$$\mathcal{L}(\theta, Y) \propto -\log|K_{yy}| - (\tilde{\boldsymbol{y}} - \boldsymbol{\mu} \otimes \mathbf{1}_T)^\top K_{yy}^{-1} (\tilde{\boldsymbol{y}} - \boldsymbol{\mu} \otimes \mathbf{1}_T) \quad (48)$$

$$\text{with} \quad K_{yy} = (C \otimes I_T) K_{xx} (C^\top \otimes I_T) + (R \otimes I_T) \quad (49)$$

where both $K_{yy}$ and $\boldsymbol{\mu}$ depend on model parameters $\theta$. A stable way to evaluate this is to begin with a Cholesky decomposition of $K_{xx} \in \mathbb{R}^{MT \times MT}$:

$$K_{xx} = LL^\top \quad (50)$$

and then use the Woodbury identity to rewrite the inverse of $K_{yy} \in \mathbb{R}^{NT \times NT}$ as:

$$K_{yy}^{-1} = (R^{-1} \otimes I_T) - (R^{-1} C \otimes I_T) \left( L^{-T} L^{-1} + C^\top R^{-1} C \otimes I_T \right)^{-1} (C^\top R^{-1} \otimes I_T), \quad (51)$$

which can be further transformed into:

$$K_{yy}^{-1} = (R^{-1/2} \otimes I_T) \left( I_N \otimes I_T - A^\top B^{-1} A \right) (R^{-1/2} \otimes I_T) \quad (\in \mathbb{R}^{NT \times NT}) \quad (52)$$

$$\text{where} \quad A \equiv L^\top (C^\top R^{-1/2} \otimes I_T) \quad (\in \mathbb{R}^{MT \times NT}) \quad (53)$$

$$\text{and} \quad B \equiv I_{DT} + AA^\top = I_{DT} + L^\top (C^\top R^{-1} C \otimes I_T) L \quad (\in \mathbb{R}^{MT \times MT}) \quad (54)$$

Using the matrix determinant lemma, $\log|K_{yy}^{-1}|$ can be simplified to:

$$\log|K_{yy}^{-1}| = \log|(R^{-1/2} \otimes I_T) \left[ I_T \otimes I_N - A^\top B^{-1} A \right] (R^{-1/2} \otimes I_T)| \quad (55)$$

$$= 2\log|(R^{-1/2} \otimes I_T)| + \log|\left[ I_N \otimes I_T - A^\top B^{-1} A \right]| \quad (56)$$

$$= -T\log|R| + \log|B - AA^\top| + \log|B^{-1}| \quad (57)$$

$$= -T\log|R| + \log|I_{DT}| - \log|B| \quad (58)$$

$$= -T\log|R| - \log|B| \quad (59)$$

$$= -T\log|R| - 2\log|W| \quad (60)$$

where $B = WW^\top$ is the Cholesky decomposition of matrix $B$. Thus, $\log|K_{yy}|$ is given by:

$$\log|K_{yy}| = T\log|R| + 2\log|W| \quad (61)$$

where $\log |R| = \sum_i \log R_{ii}$ and $\log |W| = \sum_i \log W_{ii}$.

Finally, to compute products of the form $K_{yy}^{-1} v$, we will need to compute $B^{-1} v'$ for some $v'$. To do this stably and efficiently, we solve two successive triangular systems via back-substitution: we first solve $W v'' = v'$ for $v''$, and then solve $W^\top z = v''$ for $z$ which returns $B^{-1} v'$.

This direct method of computing the marginal likelihood avoids loss of numerical precision by never explicitly computing any inverse. Moreover, it costs $\mathcal{O}((MT)^3)$, which is typically much less than the naive $\mathcal{O}((NT)^3)$. In Appendix F.2, we show how this cost can be further reduced to $\mathcal{O}(TMN + MT \log T)$ to enable large scale applications.

### F.2 Scalability to very large datasets

Here, we show how to reduce both the computational complexity and memory requirements of learning and inference in GP regression and GPFADS using the non-reversible kernels we have proposed. The methods outlined below hinge on the ability to perform very fast matrix-vector products with the Gram matrix, and contain a mix of well-known techniques and novel tricks to be published elsewhere. We first describe a set of methodological building blocks that can be used to scale up GP regression, and later explain how they apply to GPFADS too. Although the main text focuses on theoretical concepts and small-scale applications that do not make use of these acceleration techniques, we have implemented them to good effect.

**Fast matrix-vector products with the Gram matrix**

The full Gram matrix $K_{xx} \in \mathbb{R}^{MT \times MT}$ associated with the $M$-output covariance of Eq. 47 for a specific set of $T$ time bins is a sum of $2Q$ space-time Kronecker products of the form $A \otimes F$ where $A \in \mathbb{R}^{M \times M}$ and $F \in \mathbb{R}^{T \times T}$. This allows us to write efficient routines for matrix-vector multiplication with the Gram matrix, by exploiting the identity $(A \otimes F) \operatorname{vec}(V) = \operatorname{vec}(F^\top V A)$. When the time bins are regularly spaced on a grid, then $F$ is a Toeplitz matrix that can be embedded in a circulant matrix (see below), enabling the computation of $F^\top V$ products in $\mathcal{O}(MT \log T)$. Thus, the complexity of a $K_{xx} \operatorname{vec}(V)$ product can be reduced from the naive $\mathcal{O}(M^2 T^2)$ down to $\mathcal{O}(Q(M^2 T + MT \log(T)))$ (from now on, we will assume that $\log(T)$ dominates $M$, in which case this complexity simplifies to $\mathcal{O}(QMT \log(T))$. Importantly, as we will see below, this way of computing products allows us to lower the memory requirements by never storing the Gram matrix (not even any of its $T \times T$ blocks).

To compute fast $F^\top V$ products with any temporal Gram matrix $F \in \mathbb{R}^{T \times T}$ (e.g. associated with $f_q(\cdot)$ or $\mathcal{H}[f_q](\cdot)$ in Eq. 47), we use the fact that $F$ is symmetric Toeplitz when the $T$ time bins form a regular grid (Wilson and Nickisch, 2015):

$$
F = \begin{bmatrix}
f_0 & f_1 & \cdots & f_{T-2} & f_{T-1} \\
f_1 & f_0 & \cdots & f_{T-3} & f_{T-2} \\
\vdots & \vdots & \ddots & \vdots & \cdots \\
f_{T-2} & f_{T-3} & \cdots & f_0 & f_1 \\
f_{T-1} & f_{T-2} & \cdots & f_1 & f_0
\end{bmatrix}
\tag{62}
$$

One can embed this $T \times T$ Toeplitz matrix $F$ into a $2(T-1) \times 2(T-1)$ circulant matrix $F_c$, every column being a cyclically shifted version of the previous:

$$
F_c = \begin{bmatrix}
f_0 & f_1 & \cdots & f_{T-2} & f_{T-1} & f_{T-2} & \cdots & f_1 \\
f_1 & f_0 & \cdots & f_{T-3} & f_{T-2} & f_{T-1} & \cdots & f_2 \\
\vdots & \vdots & \cdots & \vdots & \vdots & \vdots & \cdots & \\
f_{T-2} & f_{T-3} & \cdots & f_0 & f_1 & f_2 & \cdots & f_{T-1} \\
f_{T-1} & f_{T-2} & \cdots & f_1 & f_0 & f_1 & \cdots & f_{T-2} \\
f_{T-2} & f_{T-1} & \cdots & f_2 & f_1 & f_0 & \cdots & f_{T-3} \\
\vdots & \vdots & \cdots & \vdots & \vdots & \vdots & \cdots & \\
f_1 & f_2 & \cdots & f_{T-1} & f_{T-2} & f_{T-3} & \cdots & f_0
\end{bmatrix} \triangleq \begin{bmatrix} F & S \\ S^\top & F' \end{bmatrix}.
\tag{63}
$$

Notice that all the information is contained within the first column, which we denote by $\boldsymbol{f}$. This is the only part of the matrix that needs to be stored explicitly. It is well known that products with a

circulant matrix can be performed in the Fourier domain as follows:

$$F_c \boldsymbol{z} = \mathrm{DFT}^{-1}\left[\mathrm{DFT}(\boldsymbol{f}) \odot \mathrm{DFT}(\boldsymbol{z})\right] \tag{64}$$

where DFT refers to the discrete Fourier transform and $\odot$ denotes element-wise multiplication. Thus, a product $F\boldsymbol{v}$ can be computed by padding the vector $\boldsymbol{v}$ with zeroes to size $2(T-1)$, then computing

$$F_c \begin{bmatrix} \boldsymbol{v} \\ \boldsymbol{0} \end{bmatrix} = \begin{bmatrix} F\boldsymbol{v} \\ S^\top \boldsymbol{v} \end{bmatrix} \tag{65}$$

and simply discarding the lower $T-2$ elements. The DFT and inverse DFT of a $T \times 1$ vector can be computed efficiently in $\mathcal{O}(T \log T)$ using the Fast Fourier transform (FFT). Thus $C\boldsymbol{z}$ can be used to exactly compute $K\boldsymbol{w}$ for any $\boldsymbol{w}$ in $\mathcal{O}(T \log T)$ computations and $\mathcal{O}(T)$ memory.

**Fast and low-memory evaluation of the marginal likelihood and its gradient**

Evaluating and differentiating the marginal likelihood in GP regression involves solving linear systems of the form $K_{xx}\boldsymbol{z} = \boldsymbol{b}$ for $\boldsymbol{z}$, as well as computing $\log |K_{xx}|$ and its gradient. Here we describe a set of old and new approaches to performing these computations at scale.

**Solving $K_{xx}\boldsymbol{z} = \boldsymbol{b}$ systems**   Computing the quadratic form in the GP log marginal likelihood, i.e. solving linear systems of the form $K_{xx}\boldsymbol{z} = \boldsymbol{b}$, can be done to numerical precision via (preconditioned) conjugate gradients (CG; Cutajar et al., 2016). This iterative method only involves $K_{xx}$ through matrix-vector products, which are fast (cf. above) and do not necessitate the explicit computation and storage of this large matrix. However, CG poses a problem when optimization is performed with the help of automatic differentiation (AD) software, an otherwise very useful way of obtaining gradients of the marginal likelihood. Naively backpropagating through every CG iteration incurs a memory cost proportional to the number of CG iterations, which is often prohibitive (in our case, the worst case would be $MT$ iterations, bringing the memory requirements close to $\mathcal{O}(M^2 T^2)$, i.e. the very cost of storing $K_{xx}$ which we seek to avoid in the first place). We were not able to find ways to circumvent this problem in the existing literature.

To mitigate the memory requirements of CG, we went back to basic AD principles and derived a novel, constant-memory way of updating the adjoint of $\boldsymbol{b}$ and $\boldsymbol{\theta}$ from the adjoint of $\boldsymbol{z} = K_{xx}(\boldsymbol{\theta})^{-1}\boldsymbol{x}$ obtained through CG iterations. We use the notation $\overline{X} = \partial \mathcal{L}/\partial X$ to denote the adjoint of $X$, where $\mathcal{L}$ is the loss (or objective) function of interest. First, recall the chain rule:

$$\overline{\theta_i} = \mathrm{Tr}\left(\overline{K_{xx}}^\top \frac{\partial K_{xx}}{\partial \theta_i}\right). \tag{66}$$

Next, for any CG solve $\boldsymbol{z} = K_{xx}^{-1}\boldsymbol{b}$ in the forward pass, the adjoint of $K_{xx}$ must be updated in the reverse pass according to $\overline{K_{xx}} \leftarrow \overline{K_{xx}} - K_{xx}^{-\top}\overline{\boldsymbol{z}}\boldsymbol{z}^\top$ (Giles, 2008). Thus, the update for $\overline{\theta_i}$ (the gradient we need for training the model) is

$$\overline{\theta_i} \leftarrow \overline{\theta_i} + \mathrm{Tr}\left[\boldsymbol{z}(-K_{xx}^{-1}\overline{\boldsymbol{z}})^\top \frac{\partial K_{xx}}{\partial \theta_i}\right]. \tag{67}$$

At this stage, although $K_{xx}^{-1}\overline{\boldsymbol{z}}$ can itself be computed in the reverse pass using CG without storing $K_{xx}$, Eq. 67 suggests that one would still need to compute and store $\partial K_{xx}/\partial \theta_i$, a matrix that is just as large as $K_{xx}$. We reasoned that automatic differentiation libraries can be used in a slightly unconventional way to evaluate Eq. 67 without ever explicitly representing neither $K_{xx}$ nor its gradient, as long as a memory-efficient matrix-vector product routine is available (which is a premise for the use of CG, anyways). The key is to note that for a matrix-vector product $\boldsymbol{d} = K_{xx}(\boldsymbol{\theta})\boldsymbol{e}$, the adjoint of $\boldsymbol{d}$ is to be propagated back to that of $\boldsymbol{\theta}$ according to (Giles, 2008):

$$\overline{\theta_i} \leftarrow \overline{\theta_i} + \mathrm{Tr}\left[\boldsymbol{e}\overline{\boldsymbol{d}}^\top \frac{\partial K_{xx}}{\partial \theta_i}\right]. \tag{68}$$

Thus, when a matrix-vector product with $K_{xx}(\boldsymbol{\theta})$ is computed under automatic differentiation, the reverse pass will automatically update the parameter vector $\boldsymbol{\theta}$ in the correct way given by Eq. 68 but without representing $\partial K_{xx}/\partial \theta_i$ explicitly. Critically, Eq. 68 has exactly the same form as Eq. 67. By identifying terms, we see that implementing Eq. 67 can be done by performing a dummy matrix-vector product $\boldsymbol{d} = K_{xx}(\boldsymbol{\theta})\boldsymbol{e}$ with $\boldsymbol{e} = \boldsymbol{z}$, and performing a reverse pass on this dummy operation,

taking care of "manually" seeding it with the adjoint $\overline{\boldsymbol{d}} = -K_{xx}^{-1}\overline{\boldsymbol{z}}$ where $\overline{\boldsymbol{z}}$ is obtained as part of the primary reverse pass. Note that this requires hijacking the primary reverse-pass on the log marginal likelihood computation, which can be done relatively easily in modern AD software (e.g. autograd, Maclaurin et al., 2015).

**Computing the log determinant and its gradient**    Exact computation of the log determinant is generally difficult for large Gram matrices. However, stochastic estimators exist for both $\log|K_{xx}|$ and its gradient that only require matrix-vector products with $K_{xx}$, again greatly alleviating the memory burden. To estimate the log determinant, we use the approach advocated by Dong et al. (2017) based on stochastic trace estimation (Filippone and Engler, 2015; Roosta-Khorasani and Ascher, 2015):

$$\log|K_{xx}| = \mathrm{Tr}(\log K_{xx}) \tag{69}$$

$$= \left\langle \boldsymbol{\xi}^\top \log(K_{xx})\boldsymbol{\xi} \right\rangle_{\boldsymbol{\xi}} \tag{70}$$

where the expectation is over any spherical distribution $p(\boldsymbol{\xi})$ with covariance equal to the identity matrix (Hutchinson's trace estimator). This expectation can be approximated with Monte Carlo samples. For computing $\log(K_{xx})\boldsymbol{\xi}$ products in Eq. 70, we wish to exploit fast and memory-efficient $K_{xx}\boldsymbol{v}$ products. One way of doing this, which we have not seen used in the GP literature, is to use the following integral representation of the matrix logarithm (Davies and Higham, 2005; Wouk, 1965):

$$\log K_{xx} = \int_0^1 (K_{xx} - I)\left[tK_{xx} + (1-t)I\right]^{-1} dt \tag{71}$$

Thus, we can estimate the log determinant of $K_{xx}$ by drawing a set of $P$ random vectors $\Xi \in \mathbb{R}^{MT \times P}$ and using any numerical quadrature method to compute

$$\log|K_{xx}| \approx \int_0^1 \mathrm{Tr}\left[Z^\top \left(tK_{xx} + (1-t)I\right)^{-1}\Xi\right] dt \qquad \text{with } Z = (K_{xx} - I)\Xi. \tag{72}$$

Here, the integrand can be computed via CG, based on matrix-vector products with $K_{xx}$. One can substantially speed up CG convergence by initializing CG iterations at a given $t$ with a previously obtained solution at a neighbouring $t'$. Note that for ill-conditioned $K_{xx}$, solvers will typically need to do more work for $t$ near $1$ – this is where adaptive solvers can help greatly.

Trace estimation also applies to the gradient of $\log|K_{xx}|$ (Cutajar et al., 2016; Dong et al., 2017):

$$\frac{\partial \log|K_{xx}|}{\partial \theta_i} = \mathrm{Tr}\left[K_{xx}^{-\top}\frac{\partial K_{xx}}{\partial \theta_i}\right] \tag{73}$$

$$= \left\langle \boldsymbol{\xi}^\top K_{xx}^{-\top}\frac{\partial K_{xx}}{\partial \theta_i}\boldsymbol{\xi} \right\rangle_{\boldsymbol{\xi}} \tag{74}$$

$$= \left\langle \mathrm{Tr}\left[\boldsymbol{\xi}\left(K_{xx}^{-1}\boldsymbol{\xi}\right)^\top \frac{\partial K_{xx}}{\partial \theta_i}\right] \right\rangle_{\boldsymbol{\xi}} \tag{75}$$

As in the computation of the quadratic form in Eq. 67, Eq. 75 seems to require an explicit representation of the large matrix $\partial K_{xx}/\partial\theta_i$. However, the same trick we introduced above can be used here too to exploit the existing machinery of AD software to evaluate Eq. 75 without storing any large matrix. Specifically, note that Eq. 75 has the exact same form as Eq. 68, such that it suffices to compute a dummy matrix-vector product $\boldsymbol{d} = K_{xx}\boldsymbol{e}$ with $\boldsymbol{e} = \boldsymbol{\xi}$ and perform a dummy reverse pass seeded with $\overline{\boldsymbol{d}} = K_{xx}^{-1}\boldsymbol{\xi}$ (obtained via CG).

In sum, these tricks reduce the complexity of estimating the model evidence and its gradient from the naive $\mathcal{O}(M^3T^3)$ to $\mathcal{O}(QMT\log T)$ (potentially with a large constant pre-factor determined by the number of CG iterations, hence the importance of good preconditioning; Cutajar et al., 2016). Similarly, the memory requirements scale as $\mathcal{O}(M^2T)$, instead of $\mathcal{O}(M^2T^2)$.

**Accelerating GPFADS**

In GPFADS, since $K_{yy} = (C \otimes I_T)K_{xx}(C^\top \otimes I_T) + R \otimes I_T$, efficient products with $K_{xx}$ also lead to efficient products with $K_{yy}$ which do not necessitate explicit storage of $K_{yy}$. Therefore, the

techniques described above apply to GPFADS directly. With $C \in \mathbb{R}^{M \times N}$, computing a $(C^\top \otimes I_T)\boldsymbol{v}$ product costs $\mathcal{O}(MNT)$, and subsequent multiplication by $K_{xx}$ costs $\mathcal{O}(QMT \log T)$ as detailed above. Thus, the cost of a $K_{yy}\boldsymbol{v}$ product – which dominates the complexity of training the GP model and computing posteriors – is $\mathcal{O}(MNT + QMT \log T)$ overall. The memory requirement is $\mathcal{O}(NT + M^2 T)$ (for small latent spaces, this is close to the cost of storing a data point in the first place).

## G  Relation to Parra and Tobar (2017)

We are only aware of one other paper introducing a non-reversible GP kernel (Parra and Tobar, 2017). Here, we outline the key differences between their models and ours, and show how our construction affords both a cleaner handle on reversibility, and better scalability properties.

Parra and Tobar (2017)'s model extended the spectral mixture model (Wilson and Adams, 2013) to the multi-output setting. In their model, the cross-spectrum of any two variables $i$ and $j$ is given by:

$$\mathbb{E}\left[ x_i(\omega)\overline{x_j(\omega)} \right] = \frac{w_{ij}}{2}\left( e^{-\frac{(\omega - \mu_{ij})^2}{2\sigma_{ij}} + j(\theta_{ij}\omega + \phi_{ij})} + e^{-\frac{(-\omega - \mu_{ij})^2}{2\sigma_{ij}} + j(-\theta_{ij}\omega + \phi_{ij})} \right), \qquad (76)$$

where $\phi_{ij}$ is a phase lag and $\theta_{ij}$ is a pure delay. The parameters $\mu_{ij}$, $\sigma_{ij}$ and $w_{ij}$ are directly constrained by the marginal spectrum of the two variables (spectral mixture) ensuring that the resulting multi-output covariance function be positive definite.

In the time domain, this leads to the following real-valued cross-covariance function:

$$K_{ij}(\tau) = w_{ij}\left( 2\pi\sigma_{ij} \right)\exp\left( -\frac{\sigma_{ij}}{2}(\tau + \theta_{ij})^2 \right)\cos((\tau + \theta_{ij})\mu_{ij} + \phi_{ij}). \qquad (77)$$

A first notable difference between this kernel and ours lies in the ability – or lack thereof – to break reversibility at low frequencies. Indeed when there are no hard delays ($\theta_{ij} = 0$), the Parra and Tobar (2017) model becomes fully reversible in the limit of a non-resonant process (one whose power spectrum has its maximum at $\omega = 0$, modeled by setting $\mu_{ij} = 0$ in Eq. 76), no matter the choice of phase lag parameters $\phi_{ij}$. Indeed, the non-reversibility index $\zeta$ is tied to the number of oscillatory cycles that fit within the envelope of the autocovariance function (Fig. 5, right). In contrast, our construction retains non-reversibility in this limit, because the Hilbert transform implies a constant phase lag at all frequencies. For example, the non-reversible squared-exponential planar covariance shown in Fig. 1 has full non-reversibility despite a complete lack of marginal oscillations in each output.

The second major difference lies in the opportunity – or lack thereof – to scale learning and inference to very large datasets. As explained in Appendix F.2, all kernels in the family that we propose take the form of a sum of space/time-separable terms, implying a specific sum-of-Kronecker-products structure for the associated Gram matrices. In contrast, Parra and Tobar (2017)'s model can never be expressed in Kronecker form, if reversibility is to be broken via the introduction of non-zero phase lags. In other words, to our knowledge, we have presented the first non-reversible multi-output kernel that can realistically support scalable learning and inference on very large datasets.