[Reviews · NeurIPS 2020]

Review 1

Summary and Contributions: The paper proposes novel Gaussian process covariance functions that model temporal non-reversibility. These covariance functions are integrated into Gaussian process factor analysis and shown to better model data produced by various dynamical systems. Performance of the method is demonstrated on both simulated dynamical systems and on real data recorded from primary motor cortex.

Strengths: The paper combines hallmarks of dynamical processes with Gaussian processes and embeds these into the popular GPFA framework. As such, it pushes the state-of-the-art in neuronal data analysis forward. The construction and incorporation of non-reversibility is elegant. The authors first introduce a measure for quantifying (second-order) non-reversibility and then use this measure to identify non-reversible components in covariance functions. They then use these elements to construct specific non-reversible covariance functions based on Hilbert transforms. The application part of the paper is convincing. They first use two dynamical systems with known ground truth to show that their method can identify sensible trajectories and then use data recorded from primary motor cortex to demonstrate that their method can reveal rotational trajectories in the latent space without providing explicit knowledge about rotations. The latter result shows that the method is useful for analyzing modern neuroscience datasets. The extensive Supplemental contains various theoretical results as well as technical considerations for efficient and numerically stable implementation of the method.

Weaknesses: The construction is useful for multi-output Gaussian processes only. This is a minor caveat given the trend to ever larger neuroscience datasets. The authors do not provide any code for their GPFADS method. I presume that code will be made available upon acceptance. Edit: The authors clarified that code will be made public on acceptance.

Correctness: I could not find any mistakes - the proposed measure for quantifying non-reversibility, the covariance function construction and the embedding into GPFA appear to be technically correct. However, various references are missing in the manuscript: lines 31, 139, 150, Figure 1 caption. I did not check all of the supplemental material for correctness.

Clarity: Clarity of the paper is outstanding. The technical part starts with an explanation of notations. There is a background section on standard GPFA which further clarifies notations. The paper then describes non-reversible Gaussian processes and their use in GPFA. The Experiments section is well structured too, first describing an application to synthetic data and then an application to real data. Figures show trajectories for the most part. They are clearly labeled, make good use of color and are generally easy to understand.

Relation to Prior Work: The paper builds on Gaussian process factor analysis and, to demonstrate the method, uses a dataset recorded from primary motor cortex that was analyzed in previous papers (Churchland et al. 2012). In the Supplemental, the authors discuss the relation to another paper that introduces non-reversible GP kernels (Parra and Tobar, 2017). To the best of my knowledge, there are no other papers on non-reversible GP kernel.

Reproducibility: Yes

Additional Feedback: The wording in lines 38-40 should be revised: the use of "directly" to describe the "indirect methods" is confusing. The discussion on the relation to Parra and Tobar, 2017 should better go into the main paper.


Review 2

Summary and Contributions: Authors provide for a novel formulation of GP based method in which underlying dynamics of the latent variables transitions are captured through the incorporation of temporal non-reversibility. The formulation allows for this to be used in a direct replacement for traditional GPFA but in the analysis of data in which the dynamic models of the latent states is of significant interest, such as in the analysis of neural population responses.

Strengths: The method is novel and very well motivated with high level of applicability across various datasets. I really appreciated that the authors took the time to verify the method on both simulated data to verify the validity and on neural population data to suggest the effectiveness of this method in uncovering interesting properties about the dataset. I’m really looking forward to seeing this method applied to highly dynamic datasets.

Weaknesses: Additional discussion on where this method could fail or would not be a good method would have been useful. EDIT: authors have addressed this minor concern and others raised by the other reviewers. Continue to highly recommend this paper for accept.

Correctness: The proof and derivations are well formulated and sound.

Clarity: Overall very well structured and clearly written. Some references within the paper were not correctly filled, thus showing up as ??

Relation to Prior Work: Relation to previous works and how this work differs from them were clearly discussed and motivated.

Reproducibility: Yes

Additional Feedback:


Review 3

Summary and Contributions: This paper defines a notion of temporal reversibility for a multi-output Gaussian process and presents a construction to generate temporarily non-reversible kernels from 1d kernels. Experiments are given on synthetic and neural data.

Strengths: This paper proposes a novel covariance construction that encodes a property that was missing from previous work. The impact of this work is potentially very high and I can envisage applications in diverse areas. For me, this was a ‘why didn’t anyone think of this before?’ sort of paper.

Weaknesses: I found the notion of reversibility quite confusing. The paper defines it as “the probability of immediately returning to an initial state must be small”, but this is not the standard definition e.g. https://en.wikipedia.org/wiki/Time_reversibility. The classical pendulum is reversible in the sense of a time reverse trajectory of a solution is a valid solution, and also all solutions are periodic and so indeed return to their initial state. What does ‘immediately returning’ mean? EDIT rebuttal clarified this for me. Score raised to 9.

Correctness: The papers appears to be correct. I verified the kernel constructions and simulated some trajectories to confirm Fig 1.

Clarity: Aside from the issue around what reversibility actually means, the paper is easy to follow and well motivated. I would be useful to have some of the content from appendix E in the main text. In particular, the phase shift of the Hilbert transform. I found this very helpful to understanding the construction.

Relation to Prior Work: This paper does not cite any work on reversibility in physical systems except in the final broader impact section. The prior work of Parra and Tobar 2017 is only briefly discussed in the main text, though the appendix F3 provides many illuminating details. I would suggest that this analysis is referenced in the main text.

Reproducibility: Yes

Additional Feedback: Typo in 256 “.however” There are few latex refs that need fixing. EDIT post rebuttal: the clarification was helpful. I have raised by a point and consider this to be an very good paper indeed.


Review 4

Summary and Contributions: This paper proposes an extension of Gaussian Process Factor Analysis (GPFA) dedicated to neural data analysis. This extension allows to account for non-reversible dynamics. The authors propose a method, based on a specific Kernel expression, to construct such non-reversible Gaussian Process. Then, they demonstrate how these GP are able to capture non-reversibility in simulated non-reversible dynamical systems. Finally, they apply the method to previously analyzed neural data from the motor cortex and show that it can recover previously observed rotational dynamics without specifically looking for it.

Strengths: * proposes a new constructive method * demonstrate how well it works on simulated trajectories * confirm that it recovers, agnostically, previously observed behavior in neural data * high-interest for comp-neuroscientists/neuroscientists as a data analysis methods needs to account for non-reversible behavior

Weaknesses: * incremental work

Correctness: * Are you sure the expression of the non-reversibility index in eq. 6 is correct ? The proof that it's between 0 and 1 in the appendix is not convincing because of a mistake or a typo or a typo in the initial definition. Expansion of eq. B.(22) is not eq. B.(23), to me there are missing terms and missing transpose (also in B. (24)-(25)). My intuition is that the integrand of the denominator in eq. 6 is ||K(tau)||^2 + ||K(-tau)||^2, otherwise, as given it is zero if K is an odd function.... * I didn't find the proof of eq. 8 in the supplementary All the rest is correct to me.

Clarity: The paper is well introduced, the methods and results are well-described. To me, it was easy to read.

Relation to Prior Work: Prior works is sufficiently cited and discussed.

Reproducibility: Yes

Additional Feedback: * Why do you call the decomposition in eq. 7 "Kronecker", any reference ? * Could you mention in the main text that the decomposition comes from a generalized SVD and give a reference for this mathematical result ? * Could you give a reference for the Heywood cases (l. 225) ? Typos and other: l. 61: isn't it vec(X)^T ? That would match the words that follow l. 102: remove "in passing", too colloquial I guess. I recommend to move this sentence after eq. 6. l. 31, l. 138, l. 150, Figure 1 caption : undefined reference ### Post author response ### Thanks for the clarification about the non-reversibility and for the reminder about the nature of K. Very nice work, I've increased my score to 9.

[Author Response · NeurIPS 2020]

We thank all reviewers for their time in reviewing our manuscript and their feedback on our work. We apologize for the
various formatting issues in the references; these are now fixed, along with typos and other linguistics mishaps [R1-4].
If accepted, we will move the discussion concerning the Parra and Tobar (2017) paper in the main text [R1, R3], as well
as the phase-shift interpretation of the Hilbert transform [R3].

**Reviewer 1 —** The authors do not provide any code for their GPFADS method. I presume that code will be made
available upon acceptance.

Yes, code will be made available online upon paper release in the form of a python library which is under preparation.

**Reviewer 2 —** Additional discussion on where this method could fail or would not be a good method would have been
useful.

Yes, we will add more discussion on the various theoretical limitations arising from the model, including the implications
of the Gaussian process assumption (second-order non-reversibility as opposed to non-reversibility in higher-order
moments; see also answer to Reviewer #3), and of the specific ways in which non-reversibility is introduced in the
kernels. Additionally, we will discuss the limitations of the simple noise model we have worked with. For single-trial
spiking data, for instance, we would expect that the model would work better if it included Poisson (as opposed to
Gaussian) observations; this is next on our list of extensions.

**Reviewer 3 —** I found the notion of reversibility quite confusing. The paper defines it as "the probabil-
ity of immediately returning to an initial state must be small", but this is not the standard definition e.g.
https://en.wikipedia.org/wiki/Time_reversibility.

We will rewrite this part of the paper to improve on clarity. Our definition of reversibility indeed follows the definition
based on detailed balance in the "Stochastic processes" section of the wikipedia page referenced by the Reviewer. We
deem a process $x(t)$ reversible if for any pair of times $t$ and $s$ and any two vectors $a$ and $b$,

$$p(x(t) = a, x(s) = b) = p(x(t) = b, x(s) = a). \tag{1}$$

If $x(t)$ is a zero-mean, stationary Gaussian process (as assumed in this paper), then it is entirely defined by its space-time
covariance function, such that the detailed balance condition above becomes a time-reversal symmetry condition for the
temporal cross-covariances. Specifically, a stationary GP is reversible if for any two time points $t$ and $s$, the covariance
matrix $\langle x(t)x(s)^T \rangle$ is symmetric.

The classical pendulum is reversible in the sense of a time reverse trajectory of a solution is a valid solution, and also all
solutions are periodic and so indeed return to their initial state. What does 'immediately returning' mean?

Thanks, we will remove this confusing definition. Concerning the pendulum, the dynamics of the angle (as an
observation) are indeed fully reversible. However, the dynamics of the system, considering its full state $(\theta, \dot{\theta})$, are highly
non-reversible: oscillations in $\theta$ arise from near-circular state trajectories in the $(\theta, \dot{\theta})$ plane that evolve clockwise, but
never counter-clockwise.

**Reviewer 4 —** Are you sure the expression of the non-reversibility index in Eq. 6 is correct ? [. . . ] Expansion of Eq.
B.(22) is not Eq. B.(23) [. . . ] My intuition is [. . . ], otherwise, as given it is zero if $K$ is an odd function.

We have doubled checked, and Eq. 6 is indeed correct. The expression is simplified using the fact that (by stationarity)
$K(-\tau) = K(\tau)^T$ – we will add this point to the paragraph preceding the equation. (Also, just to clarify, $K(\cdot)$ is a
covariance function and can never be odd.)

I didn't find the proof of Eq. 8 in the supplementary

Thank you for pointing out this oversight, this will be added. In short, Eq. 7 is an orthogonal decomposition, such that
the sum of squares in $K(\cdot)$ (as a matrix-valued function) is equal to the sum of squared weights in the decomposition
(i.e. the sum of $\lambda^2$). Moreover, since the two sums in Eq. 7 separately decompose the numerator and denominator in Eq.
6 (uniqueness of the symmetric/skew-symmetric decomposition of a matrix-valued function), Eq. 8 follows.

Why do you call the decomposition in Eq. 7 "Kronecker", any reference?

Equation 7 defined the space-time covariance $K(\tau)$ as a function of the time-lag $\tau$. Since each term in the sum is the
product of a spatial component and a temporal component, any Gram matrix instantiating the kernel at a discrete set of
time points is a sum of Kronecker products. We will explain the origin of this terminology in the main text.

Could you mention in the main text that the decomposition comes from a generalized SVD and give a reference for this
mathematical result? Could you give a reference for the Heywood cases (l. 225)?

We will add references for these in the text. These will include: C. Van Loan, Journal of comp. and applied mathematics.
(2000), Crane et al, SIAM J. Numer. Anal. (2020) and Martin, J.K. et al, Psychometrika 40, 505-517, (1975).

[Meta-Review · NeurIPS 2020]

Four knowledgeable referees support accept and I accept. Several typos in the paper need to be corrected (for instance, replace ? for the adequate references).